# Learning Efficient Models From Few Labels By Distillation From Multiple Tasks

## Abstract

We address the challenge of getting efficient yet accurate recognition systems that can be trained with limited labels. Many specialized applications of computer vision (e.g. analyzing X-rays or satellite images) have severe resource constraints both during training and inference. While transfer learning is an effective solution for training on small labeled datasets it still often requires a large base model for fine-tuning. In this paper we present a weighted multi-source distillation method; we distill multiple (diverse) source models trained on different domains, weighted by their relevance for the target task, into a single efficient model using limited labeled data. When the goal is accurate recognition under computational constraints, our approach outperforms both transfer learning from strong ImageNet initializations as well as state-of-the-art semi-supervised techniques such as Fix-Match. When averaged over 8 diverse target tasks our method outperform the baselines by 5.6%-points and 4.5%-points, respectively.

## 1 Introduction

With recent advances in recognition, there is an increasing interest in deploying deep networks in a variety of downstream applications, be it analyzing X-rays, skin conditions, or satellite images. However, in contrast to the increasingly massive training datasets and large networks that power advances in deep learning, many of these downstream applications have severe resource constraints. In particular, *labeled training data is expensive*. In addition, we often require *fast and cheap inference* as privacy and availability concerns, as well as practical limitations often require deployment to be on local devices. These constraints immediately rule out the standard approach of training a large neural network on massive amounts of training data. A key research question for these applications is thus: how do we get efficient but accurate recognition systems that we can train with limited labels?

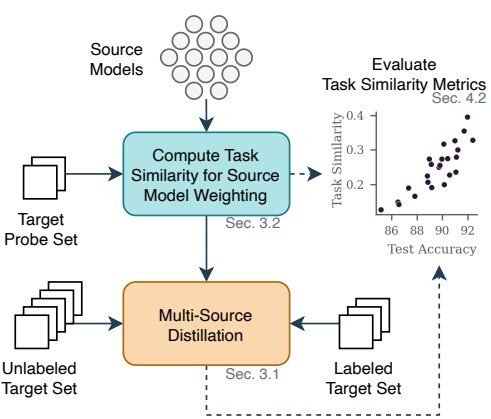

Figure 1: Overview of our paper and method.

A common approach for dealing with small labeled datasets is transfer learning. Here, one pre-trains a base model on a different problem where large labeled datasets are available, and then fine-tunes this model on the application of interest. This transferred model will often be large, but can then be *distilled* into a smaller model for deployment (Ba & Caruana, 2014; Hinton et al., 2015). While in principle this approach can be effective, in practice this relies heavily on having a good source model that is relevant to the target task. There are many ways to compare and choose the best base model (Achille et al., 2019; Kornblith et al., 2019b; Recht et al., 2019; Bolya et al., 2021). But this assumes that a *single* optimal base model exists. What if no single source model matches the target task, as is likely going to be the case for new problem domains?

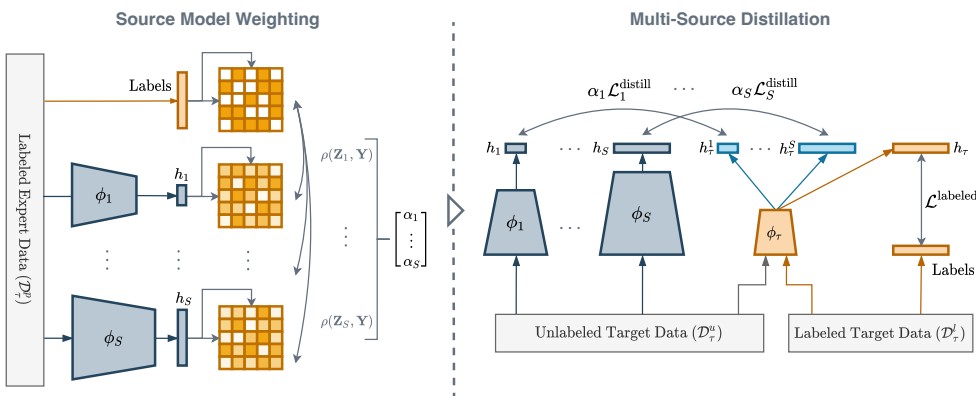

Figure 2: While common multi-source distillation usually weighs a set of $S$ source models, $\mathcal{M}_s = h_s \circ \phi_s$, equally for every target task, we propose to weight the source models by using task similarity metrics to estimate the alignment of each source model with the particular target task using a small subset of labeled data, $\mathcal{D}_\tau^p$. Since the task similarity metrics are independent of feature dimension, we can utilize source models of any architecture and from any source task. We show that choosing the weighting, $\alpha_1, \ldots, \alpha_S$, this way we are able to improve performance over transfer from ImageNet and training with FixMatch (see e.g. Table 1 and Figure 3).

This challenge motivates our approach. We propose to produce efficient yet accurate models for new tasks with few labels by *task-similarity-weighted multi-source distillation* (see Figure 2). That is, we *distill* multiple (diverse) source models trained on different domains, weighted by their relevance for the target task. All this is done without access to any other data than that of the target task. Specifically, we propose to first rank a diverse set (both in architecture and task) of source models for a particular target task using a task similarity metric. This ranking is used to select and weight the most relevant source models to distill into a target model of some suitable target-architecture in a semi-supervised learning setting using multi-source distillation (see Figure 1).

**Contributions.** We summarize our contributions as follows: 1) By analyzing over 200 distilled models we extensively verify that, for single-source cross-domain distillation, the choice of source model is important for the predictive performance of the target model. 2) We show that *task similarity metrics* can be used to select and weight source models for single- and multi-source distillation without access to any source data. 3) We show that our approach yields the best accuracy on multiple target tasks under compute and data constraints. We compare our task similarity-weighted multi-source distillation to two baselines; classical transfer learning and FixMatch, as well as the special case of multi-source distillation with *equal* weighting. Averaged over 8 diverse datasets, our method outperforms the baselines with at least 4.5%-points and in particular 17.5%-points on CUB200.

## 2 RELATED WORK

We review several research areas relevant to our problem setup and approach.

**Knowledge Distillation** One key aspect of our problem is to figure out how to compress multiple models into an efficient target model. A common approach is knowledge distillation (Ba & Caruana, 2014; Hinton et al., 2015) where an efficient student model is trained to mimic the output of a larger teacher model. However, most single-teacher (Adriana et al., 2015; Mirzadeh et al., 2019; Park et al., 2019; Cho & Hariharan, 2019; Borup & Andersen, 2021) or multi-teacher knowledge distillation (You et al., 2017; Fukuda et al., 2017; Tan et al., 2019; Liu et al., 2020) research focuses on the closed set setup, where the teacher(s) and the student both attempts to tackle the same task. To the best of our knowledge, compressing multiple models specializing in various tasks different from the target task has rarely been explored in the literature. Our paper explores this setup and illustrate that carefully distilling multiple source models can bring forth efficient yet accurate models.

**Semi-Supervised Learning and Semi-Supervised Transfer**    Given our target task is specified in a semi-supervised setting, it is customary to review semi-supervised learning (SSL). The key to SSL approaches is how to effectively propagate label information from a small labeled dataset to a large unlabeled dataset. Along this vein, methods such as pseudo-labeling/self-training (Lee et al., 2013; Xie et al., 2020) or consistency regularization (Tarvainen & Valpola, 2017; Berthelot et al., 2019; Sohn et al., 2020) have shown remarkable results in reducing deep networks dependencies on large labeled datasets via unlabeled data. However, most SSL approaches focus on training models from scratch without considering the availability of pre-trained models. Given the increasing availability of large pre-trained models (Paszke et al., 2019; Wolf et al., 2019), recent work has started exploring the intersection between transfer learning and SSL (Phoo & Hariharan, 2021; Islam et al., 2021; Abuduweili et al., 2021). However, most of these works focus on how to transfer from a single pre-trained model to the target task. Our paper, however, explores an even more practical setup: how to transfer from multiple pre-trained models to a downstream task where in-domain unlabeled data are available. In principle, we could combine our approach with a lot of previous work on SSL to (potentially) gain even larger improvements, but to keep our method simple we leave such exploration to future work and focus on how to better utilize the available set of pre-trained models.

**Multi-Source Domain Adaptation**    Our setup also bears a resemblance with multi-source domain adaptation (MSDA) (Peng et al., 2019) in which the goal is to create a target model by leveraging multiple source models. However, MSDA methods often assume that the source and target models share the same label space to perform domain alignment. We do not make such an assumption and in fact, focus on the case where the label space for source and target tasks has minimal to no overlap. Besides, a lot of the MSDA approaches (Zhao et al., 2018; Xu et al., 2018; Peng et al., 2019; Zhao et al., 2020) rely on the availability of source data or the fact that the source and target tasks share the same model architecture to build domain invariant features. Given the discrepancy in assumptions between multi-source domain adaptation and our setup, we do not consider any methods from this line of work as baselines.

**Transfer Learning From Multiple Sources**    Transfer learning from multiple different pre-trained models has been explored in different setups. Bolya et al. (2021) focuses on how to select a single good pre-trained model to use as a model initialization whereas we explore how to construct an efficient model from the pre-trained models (i.e. our target architecture could be different from those of the source models). Agostinelli et al. (2022) focuses on how to select a subset of pre-trained models to construct an (fine-tuned) ensemble, whereas we focus on creating a single model that could potentially have different architecture complying with computational constraints. Li et al. (2021) focuses on creating a generalist representation by distilling multiple pre-trained models using proxy/source data (which often requires high capacity models) whereas our goal is to construct an efficient specialist model using the target data. All these works have indicated the importance of exploring how to best leverage a large collection of pre-trained models but due to differences in setup and assumptions, we do not (and could not) compare to them.

**Task Similarity / Transferability Metrics**    The key insight of our approach is to leverage the similarity between the target and source tasks to weigh different pre-trained source models during distillation. Characterizing tasks (or similarities between tasks) is an open research question that has had various successes. A common approach is to embed tasks into a common vector space and characterize similarities between tasks by distances in said space. Representative research along this line of work include Achille et al. (2019); Peng et al. (2020); Wallace et al. (2021). Another related line of work investigates transferability metrics (Tran et al., 2019; Bao et al., 2019; Nguyen et al., 2020; Dwivedi et al., 2020; Dwivedi & Roig, 2019; Bolya et al., 2021). After all, one of the biggest use cases of task similarities is to predict how well a model transfers to new tasks (models trained on similar tasks likely are good candidates for transfer learning). Since it is not our intention to establish which task similarity/transferability metric is the best for distillation, we use established metrics that capture the similarity between source representations and one-hot labels to weigh the source models. Under this purview, metrics that characterize similarities between features such as CKA (Cortes et al., 2012; Kornblith et al., 2019a) and transferability metrics based on features (Dwivedi & Roig, 2019; Bolya et al., 2021) fits the bill. We evaluate the correlation between these metrics and distillation performance in Section 4.2.1.

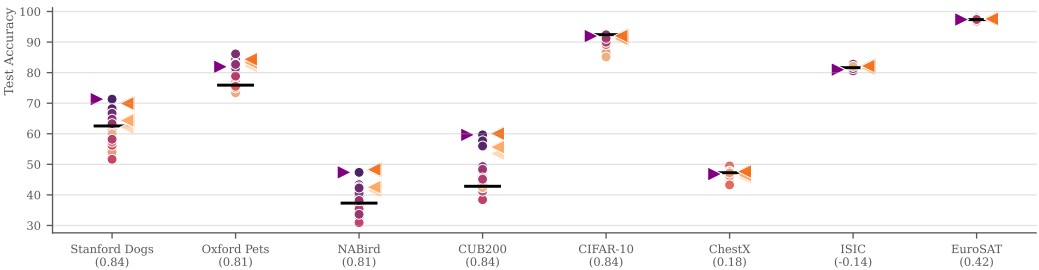

Figure 3: Distillation results for multiple target tasks with each dot representing the test accuracy after distillation from a source model. The coloring represents the task similarity associated with the particular source model (from small to large; ▬). We include the performance when fine-tuning ImageNet (▬), distilling the source model highest ranked by task similarity (▶), and performing multi-source distillation with equal weights (◀), and weights proportional to task similarity at power 1 (◀), and 12 (◀), respectively. The numbers in parentheses are Spearman correlations between the task similarity and test accuracy for the single-source distilled models of each target task.

# 3 PROBLEM SETUP AND METHOD

We want to train an accurate model for a target task, where we have limited labeled data and computational constraints (e.g. limited compute resources/storage) during inference. As discussed above, we want to leverage various pre-trained models (i.e. different architectures trained on different pre-training tasks). Formally, we assume that our target task is specified via a small labeled training set $D_\tau^l$. To mitigate the data problem, we propose to leverage: (a) a set $\mathcal{S} = \{\mathcal{M}_s\}_{s=1}^{S}$ of $S$ *source* models $\mathcal{M}_s$ trained on various source tasks different from the target task and (b) a set of unlabeled data $D_\tau^u$ associated to the target task.

To reflect computational constraints, we assume that the architecture of the target model $\mathcal{M}_\tau$ must be chosen to fit these constraints. This might mean that the architecture we need may not be available among the source models in $\mathcal{S}$. We also assume no access to any source data which could be practical due to storage, privacy, and computational constraints.

## 3.1 MULTI-SOURCE DISTILLATION

Since the target model might not share the same architecture with any of the source models, we propose to train our target model by *distilling* from multiple source models, with each source model weighted based on its relevance for the target task (See Figure 2). For simplicity, we restrict our models (regardless of source or target) to classification models that could be parameterized as $\mathcal{M} = h \circ \phi$ (the feature extractor $\phi$ embeds input $\mathbf{x}$ into a feature representation and the classifier head, $h$, maps the feature $\phi(\mathbf{x})$ into predicted class probabilities, $P(\mathbf{y} \mid \mathbf{x})$).

To train the target model $\mathcal{M}_\tau = h_\tau \circ \phi_\tau$, we utilize a weighted sum of two loss functions. The first loss function is the standard supervised objective,

$$\mathcal{L}^{\text{labeled}} \stackrel{\text{def}}{=} \frac{1}{N_l} \sum_{(\mathbf{x}_i, \mathbf{y}_i) \in \mathcal{D}_\tau^l} \ell_{CE}\left(h_\tau(\phi_\tau(\mathbf{x}_i)), \mathbf{y}_i\right),$$

where $\ell_{CE}(\cdot, \cdot)$ is the classical cross-entropy loss. The second loss function consists of a distillation objective for each source model $\mathcal{M}_s = h_s \circ \phi_s$. Specifically, for each source model (indexed by $s$) the *distillation objective* is

$$\mathcal{L}_s^{\text{distill}} \stackrel{\text{def}}{=} \frac{1}{N_u} \sum_{\mathbf{x}_i \in \mathcal{D}_\tau^u} \ell_{CE}\left(h_\tau^s(\phi_\tau(\mathbf{x}_i)), h_s(\phi_s(\mathbf{x}_i))\right) \quad \text{for } s = 1, \ldots, S,$$

where $h_\tau^s$ is an additional classifier head that maps the features from the target task feature extractor $\phi_\tau$ to the label space of the source task (to account for the difference in label space). Note, we discard all these classifier heads $h_\tau^s$ after training and merely keep the target classifier head $h_\tau$.

Combined, the overall objective function is as follows,

$$\mathcal{L} \stackrel{\text{def}}{=} \lambda \mathcal{L}^{\text{labeled}} + (1 - \lambda) \sum_{s=1}^{S} \alpha_s \mathcal{L}_s^{\text{distill}},$$

where $\lambda, \alpha_1, \ldots, \alpha_S \in [0, 1]$. Here $\alpha_s$ is the relative weight assigned to each source model such that $\sum_{s=1}^{S} \alpha_s = 1$. In principle, we could add additional semi-supervised losses, such as the FixMatch loss (Sohn et al., 2020) to propagate label information from the labeled set $\mathcal{D}_\tau^l$ to the unlabeled set $\mathcal{D}_\tau^u$ for better performance, but we avoid such additional complexities (e.g hyperparameters) and confounding the effect of distilling from multiple models. We leave such exploration to future work.

## 3.2 TASK RELEVANCE WEIGHTING FOR MULTI-SOURCE DISTILLATION

Simply assigning equal weight (i.e. $\alpha_1 = \cdots = \alpha_S = 1/S$) to all source models is usually sub-optimal (e.g. weighing source models trained on ImageNet and Chest X-ray equally might not be optimal for recognizing birds). As such, we propose to compute the source weights $\{\alpha_i\}_{i \in [S]}$ as

$$\alpha_i = \frac{\underline{e}_i^p}{\sum_{s=1}^{S} \underline{e}_s^p}, \quad \text{where} \quad \underline{e}_j = \mathbb{1}_{(e_j > 0)} \, e_j \quad \text{for } j = 1, \ldots, S,$$

where $e_s$ is a similarity between the $s$-th source tasks and the target task, and $p$ is a hyperparameter used to rescale the distribution of the weights. Larger $p$ would assign more weight to the most relevant source models, while $p = 0$ corresponds to equal weight and $p \to \infty$ assigns all weight to the single most relevant source model.

Quantifying similarities between tasks is an open research question with various successes (Achille et al., 2019; Nguyen et al., 2020). For simplicity, we pick our similarity based on one simple intuition: target examples with the same one-hot label should have similar source representations and vice versa. Along this vein, the recently introduced metric, PARC, fits the bill (Bolya et al., 2021).

For convenience, we briefly review PARC. Given a small labeled probe set $\mathcal{D}_\tau^p = \{(\mathbf{x}_i, \mathbf{y}_i)\}_{i=1}^n$ and a source representation of interest $\phi_s$, PARC first constructs two distance matrices $D_{\phi_s}, D_Y$ based on the Pearson correlations between every pair of examples in the probe set;

$$D_{\phi_s} = 1 - \text{pearson}(\{\phi_s(\mathbf{x}_i)\}_{i=1}^n), \quad D_Y = 1 - \text{pearson}(\{\mathbf{y}_i\}_{i=1}^n).$$

PARC is computed as the Spearman correlation between the lower triangles of the distance matrices;

$$e_s = \text{PARC}(\phi_s, Y) = \text{spearman}\left(\{D_{\phi_s}[i, j]\}_{i<j}, \{D_Y[i, j]\}_{i<j}\right).$$

Intuitively, PARC quantifies the similarity of representations via comparing the (dis)similarity structures of examples within different feature spaces: if two representations are similar, then (dis)similar examples in one feature space should stay (dis)similar in another feature space. In Figure 3 we show how ranking source models by PARC correlates well with distillation accuracy, and that selecting an appropriate source model can yield significant improvements.

## 4 EXPERIMENTS AND RESULTS

### 4.1 EXPERIMENTAL SETUP

**Benchmark.** For our setting, we modify an existing transfer learning benchmark: Scalable Diverse Model Selection (SDMS) by Bolya et al. (2021). In particular, we used the publicly available models to construct the set of source models for each target task. This full set consists 32 models[1]; 4 architectures (AlexNet, GoogLeNet, ResNet-18, and ResNet-50 (Krizhevsky et al., 2012; He et al., 2016)) trained on 8 different tasks. For the target tasks, we consider 8 different tasks covering various image domains (Natural images: CIFAR-10, CUB200, NABird, Oxford Pets, Stanford Dogs; X-ray: ChestX; Skin Lesion Images: ISIC; Satellite Images: EuroSAT). For the target architecture, we use MobileNetV3 (Howard et al., 2019) due to its lower computational requirements compared to any of the source models. We refer the reader to the supplementary material for further details on

---

[1]We carefully leave out any source models associated with the target task if such exists.

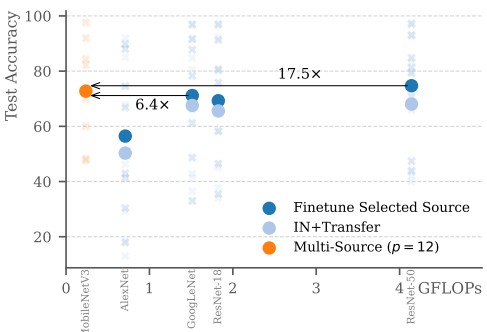 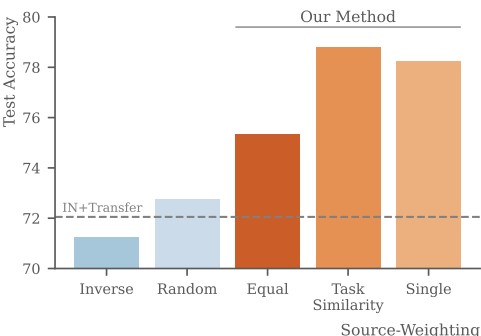

Figure 4: Mean test accuracy over the 8 target tasks compared to compute requirements for a single forward pass at inference. Using multi-source distillation allows for training of significantly smaller models with small to no loss in predictive performance.

Figure 5: Average test accuracy over five target tasks for different source-weighting methods: INVERSE is TASK SIMILARITY weights applied to a reversed sorted list of source models, while RANDOM is a sampled weight-vector on the uniform 4-Simplex. When applicable $p = 12$.

implementation. We also experimented with VTAB (Zhai et al., 2019) where we observe similar results on the natural and specialized tasks. Results on VTAB can be found in Appendix A.

**Baselines.** To show the importance of selecting the right source model to distill, we consider two baselines based on ImageNet initializations: IN+TRANSFER fine-tunes ImageNet representations using the labeled data, and IN+FIXMATCH fine-tunes the ImageNet representation using labeled and unlabeled data (via FixMatch). We also consider distilling all source models with equal weights to show the importance of using task similarity-weighting. Additionally, we report the expected performance if we were to pick a single source model at random (SINGLE SOURCE (AVG.)).

### 4.1.1 RESULTS

We present our main results in Figure 3 with exact (and additional) values reported in Table 1.

**Distillation is better than fine-tuning from ImageNet.** We observe that within the same target architecture (MobileNetv3), simply fine-tuning ImageNet representations (IN+TRANSFER) is less optimal than distilling from the most similar SINGLE-SOURCE model (selected with our task similarity metric). In fact, for fine-grained datasets such as CUB200, NABird, Oxford Pets, and Stanford Dogs, we observe that distilling from an appropriate source model (SINGLE-SOURCE) could yield much better performance than fine-tuning from a generalist ImageNet representation. More surprisingly, even with the aid of unlabeled data, models fine-tuned from ImageNet representations using a label propagation style approach (IN+FIXMATCH) still underperform distillation-based methods by at least 3.9% on average (Table 1).

**Distilling all sources equally is not always optimal.** Compared to distilling from the most similar source (SINGLE-SOURCE) or from a set of sources weighted by task similarity (MULTI-SOURCE $p = 12$), distilling all source models equally is far less optimal (see also Figure 5). This vindicates the importance of distillation using weighted source models and accentuates that distilling additional irrelevant source models can decrease performance.

**Distilling from multiple source models can outperform distilling a single best source model.** On Oxford Pets (classification of different breeds of cats and dogs), we observe that distilling from multiple weighted sources (MULTI-SOURCE $p = 12$) is much better than distilling from the single most similar source (SINGLE-SOURCE), which is a ResNet-18 trained on Caltech101 (that can recognize visual concepts such as Dalmatian dog, spotted cats, cougars, etc.). Although the most similar source model contains relevant information for recognizing different breeds of dogs and cats, it might not contain all relevant knowledge from the set of source models that could be conducive

Table 1: Multi-source distillation compared to baselines. Despite requiring fewer computations, MobileNetV3 models trained with our method are highly competitive with baseline methods for more demanding model architectures. We highlight the top 3 methods, which comply with compute requirements (i.e. MobileNetV3) for each target task by **bold**, underline, and *italic*, respectively.

| | | CIFAR-10 | CUB200 | ChestX | EuroSAT | ISIC | NABird | Oxford Pets | Stanford Dogs | Mean |
|---|---|---|---|---|---|---|---|---|---|---|
| AlexNet (0.71 GFLOPs) | IN+Transfer | 85.0 | 18.4 | 46.2 | 91.9 | 67.8 | 13.0 | 50.9 | 29.1 | 50.3 |
| | Fine-tune Selected Source | 88.0 | 30.4 | 42.9 | 89.8 | 74.5 | 17.9 | 66.8 | 41.3 | 56.5 |
| GoogLeNet (1.51 GFLOPs) | IN+Transfer | 91.8 | 42.8 | 41.4 | 96.8 | 80.5 | 36.5 | 84.8 | 65.9 | 67.6 |
| | Fine-tune Selected Source | 91.6 | 61.2 | 48.6 | 96.9 | 78.3 | 33.0 | 87.8 | 71.8 | 71.2 |
| ResNet-18 (1.83 GFLOPs) | IN+Transfer | 92.2 | 37.8 | 45.2 | 96.6 | 80.2 | 34.0 | 80.2 | 58.2 | 65.6 |
| | Fine-tune Selected Source | 91.3 | 58.2 | 46.4 | 97.0 | 75.8 | 35.4 | 80.7 | 69.3 | 69.3 |
| ResNet-50 (4.14 GFLOPs) | IN+Transfer | 92.9 | 42.0 | 43.4 | 96.8 | 79.9 | 39.9 | 83.3 | 65.9 | 68.0 |
| | Fine-tune Selected Source | 93.0 | 70.8 | 43.9 | 97.2 | 81.3 | 47.4 | 84.8 | 79.3 | 74.7 |
| MobileNetV3 (0.24 GFLOPs) | IN+Transfer | 92.4 | 42.8 | 47.3 | 97.4 | 81.6 | 37.3 | 75.9 | 62.6 | 67.2 |
| | IN+FixMatch | **93.5** | 41.9 | 38.5 | **98.1** | **82.6** | *42.8* | 83.4 | 65.8 | 68.3 |
| | Single-Source (Avg.) | 89.6 | 46.5 | 46.6 | 97.4 | *81.8* | 39.0 | 79.4 | 61.9 | 67.8 |
| | **(Ours)** Multi-Source (Equal) | 90.8 | *53.5* | 45.7 | 97.5 | 81.5 | 41.4 | *82.1* | 62.1 | *69.3* |
| | **(Ours)** Single-Source ($p \to \infty$) | 92.0 | 59.6 | *46.8* | 97.4 | 81.0 | 47.4 | 81.9 | **71.3** | 72.2 |
| | **(Ours)** Multi-Source ($p = 12$) | *92.0* | **60.0** | **47.7** | 97.6 | 82.2 | **48.3** | **84.4** | 69.9 | **72.8** |

to recognizing all visual concepts in Oxford Pets. In fact, we observe that the second most similar model is a GoogLeNet model trained on Stanford Dogs that could recognize more different breeds of dogs than the most similar source model (but incapable of recognizing cats). By distilling from multiple weighted sources, we are able to effectively combine knowledge from different source models for a more accurate target model compared to a target model distilled from a single source.

**Distilling to efficient architecture could be better than fine-tuning large models.** In Figure 4 (and also Table 1), we include the performance when fine-tuning larger architectures trained on ImageNet and the source model (of same architecture) most similar to each target task. A few observations is immediate: (a) our choice of task similarity metric is effective for transfer; across all 4 architectures, we observe at least 4% improvement over simple fine-tuning from ImageNet, which validates the results by Bolya et al. (2021), and (b) with the aid of unlabeled data and distillation, the computationally efficient architecture MobileNetV3 can outperform larger architectures fine-tuned on labeled data from the target task (i.e. AlexNet, GoogLeNet, ResNet-18). Although underperforming fine-tuning a ResNet-50 initialized with the most similar (ResNet-50) source model by a mere average of 2%-points (FINE-TUNE SELECTED SOURCE), using a ResNet-50 would require $17.5\times$ more computations during inference to achieve such improvements.

## 4.2 ABLATIONS

### 4.2.1 TASK SIMILARITY FOR DISTILLATION

There are many existing metrics for quantifying task similarities but their effectiveness for distillation remains unclear. Given the myriads of metrics, we restrict our focus on metrics that can capture similarities between a source representation of a target example and its one-hot label representation. Along this vein, two questions arise: which metric to use for comparing representations and which representations from a source model should be used to represent a target example?

For the first question, we look into multiple metrics in the literature that compares various representations: CKA (Cortes et al., 2012), RSA (Dwivedi & Roig, 2019), and PARC (Bolya et al., 2021); for the second question, we look into the common representations from a source model: the feature $\phi$ and the probabilistic output $h \circ \phi$. For simplicity, we focus on single-source distillation (i.e. how to pick the best source model for distillation).

Table 2: Spearman correlation between test accuracy after all possible single-source distillations and task similarities associated with the source models. We do not use any heuristic as e.g. in Bolya et al. (2021). Generally feature representations correlate better with distillation performance compared to pseudo-label representations.

| | | CIFAR-10 | CUB200 | ChestX | EuroSAT | ISIC | NABird | Oxford Pets | Stanford Dogs | Mean |
|---|---|---|---|---|---|---|---|---|---|---|
| Pseudo | CKA | 0.72 | 0.62 | 0.23 | 0.39 | -0.04 | 0.31 | 0.69 | 0.11 | 0.38 |
| | PARC | 0.79 | 0.79 | 0.02 | 0.17 | 0.06 | 0.48 | 0.72 | 0.54 | 0.45 |
| | RSA | 0.82 | 0.31 | -0.11 | 0.30 | **0.10** | -0.03 | 0.65 | 0.38 | 0.30 |
| Feature | CKA | 0.82 | 0.39 | **0.36** | 0.21 | -0.04 | 0.47 | 0.69 | 0.55 | 0.43 |
| | PARC | 0.84 | **0.84** | 0.18 | **0.42** | -0.14 | **0.81** | 0.81 | 0.84 | **0.58** |
| | RSA | **0.86** | 0.81 | 0.03 | 0.38 | 0.03 | 0.28 | **0.89** | **0.85** | 0.52 |

Table 3: Relative accuracy of top-3 single-source distilled target models selected by task similarity over the average of the 3 best models found in hindsight. We compute the average test accuracy of the top-3 highest ranked target models (ranked by some task similarity) and divide this average with the average of the test accuracy of the 3 best performing target models.

| | | CIFAR-10 | CUB200 | ChestX | EuroSAT | ISIC | NABird | Oxford Pets | Stanford Dogs | Mean |
|---|---|---|---|---|---|---|---|---|---|---|
| Pseudo | CKA | 99.1 | 95.6 | 97.4 | 99.6 | 98.8 | 89.4 | **100.0** | 97.6 | 97.2 |
| | PARC | 99.5 | **100.0** | 95.5 | 99.6 | 98.5 | 99.7 | 98.8 | **99.7** | 98.9 |
| | RSA | **100.0** | 77.7 | 96.5 | 99.7 | 98.5 | 87.2 | 98.6 | 97.6 | 94.5 |
| Feature | CKA | **100.0** | 95.6 | 97.0 | **99.8** | **99.0** | 93.3 | **100.0** | 96.4 | 97.6 |
| | PARC | **100.0** | **100.0** | **97.8** | 99.7 | 98.3 | **100.0** | 97.1 | 98.5 | **98.9** |
| | RSA | **100.0** | **100.0** | 96.7 | 99.8 | 98.9 | 94.9 | 98.9 | 98.8 | 98.5 |

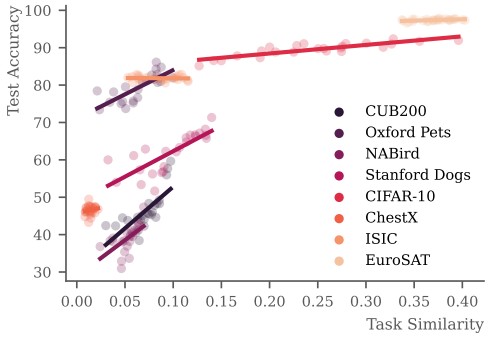

Figure 6: Test accuracy of single-source distillation and (unnormalized) task similarity score calculated using PARC on the feature representations. The scores are on different scales for different tasks, but all tasks (except ISIC) have a positive Spearman and Pearson correlation between test accuracy and task similarity.

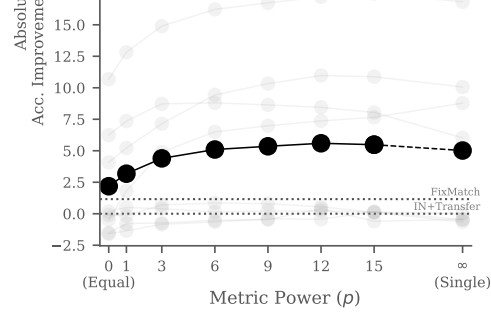

Figure 7: Improvement over transfer baseline. Here ● is the mean improvement across all target tasks and each ● represents the performance on a target dataset with given $p$. Note, $p = 0$ corresponds to equal weight for all source models, and $p = \infty$ the single highest ranked model.

To establish the effectiveness of our choice of similarity metric, we report the Spearman correlation between the task similarities and the test accuracy of the distilled models in Table 2. We see that features from the source models can better capture the correlation between the source models and the test accuracy of the distilled models, than the probabilistic pseudo-labels. In addition, we also see a much higher correlation among natural tasks (compared to specialized tasks such as ChestX, EuroSAT, and ISIC) which suggests that our choice of task similarity is effective at selecting similar tasks. Besides, we also observe a much higher correlation when using PARC compared to the other metric, thus validating our choice of using PARC as the default metric.

To further establish that our choice of metric can be used to rank various source models, we look at the relative test accuracy between the top-3 models most similar to the target task and the top-3 best models after performing distillation (see Table 3). Again, we observe that all of the three metrics are capable of ranking affinity between source models, but ranking the models with PARC outperforms the other two metrics.

### 4.2.2 Should you weigh based on task similarity?

We have established that our choice of similarity metric can capture the correlation between the source model representations and the test accuracy of the distilled models. However, it is not a prior clear that weighing different sources based on the ranking of their affinity to the target task would yield better performance for multi-source distillation. As such, we investigate different choices of weighing schemes for a subset of 5 target tasks across different domains (CUB200, EuroSAT, ISIC, Oxford Pets, Stanford Dogs): INVERSE (source model weights are inversely proportional to task similarity), RANDOM (weights are samples from a 4-simplex), EQUAL (setting equal weights for all source models), TASK SIMILARITY (default), and SINGLE (distilling from the most similar source).

Through Figure 5, we find that distilling from a single or set of source models ranked using the similarity metric (SINGLE, TASK SIMILARITY) is much more effective than distilling from source models that are weighted randomly or equally (RANDOM or EQUAL). In addition, the fact that INVERSE underperforms IN+TRANSFER on average suggests that it is crucial to follow the ranking induced by the similarity metrics when distilling the sources and that the metric ranks not only the most similar source models appropriately, but also the least similar source models.

### 4.2.3 What is the effect of p?

Our task similarity metric gives a good ranking of which source models to select for distillation but it is unclear whether the similarity score could be used directly without any post-processing. To investigate, we start by visualizing the relationship between the test accuracy of the models distilled from a single source and our task similarity. From Figure 6, it is clear that the distribution of task similarities is dependent on the target task. This observation motivates our normalization scheme.

In addition, it is not apriori clear that the appropriate weights should scale linearly with respect to the similarity scores. Thus, we investigate the effect of the rescaling factor, $p$, for constructing the weights for distillation. In Figure 7, we see that although no rescaling ($p = 1$) outperforms equal weighting ($p = 0$), it is far less optimal than rescaling with $p = 12$ (our default). This suggests that the relationship between our task similarity and good weights is monotonic but non-linear.

### 4.3 Additional Ablations and Results

We include additional ablations and results in Appendix A, but summarize these findings as follows.

- MULTI-SOURCE distillation with ResNet-50 as target model; averaged over 8 tasks, task similarity-weighted multi-source distillation outperforms both IN+TRANSFER and equally weighted multi-source distillation. Furthermore, when initializing our target model with the most similar ResNet-50 from the set of source models, performing multi-source distillation improves over multi-source distillation from an ImageNet initialization.

- Multi-source distillation on VTAB; both MULTI-SOURCE and SINGLE-SOURCE outperform IN+TRANSFER averaged over the ● *Natural* and ● *Specialized* tasks.

- Multi-source distillation with even fewer labeled samples; MULTI-SOURCE and SINGLE-SOURCE both improve in predictive performance over IN+TRANSFER, indicating our method works with even fewer labeled samples.

- Additional investigations of task similarity metrics; we consider both Spearman, Pearson, and Kendall Tau correlation as well as additional measures of relative accuracy of the top-$k$ selected models - all supporting the usefulness of task similarity to weigh source models.

## 5 Conclusion

We investigate the use of diverse source models to obtain efficient and accurate models for visual recognition with limited labeled data. In particular, we propose to distill multiple diverse source models from different domains weighted by their relevance to the target task without access to any source data. We show that under computational constraints and averaged over a diverse set of target tasks, our method outperforms both transfer learning from ImageNet initializations and state-of-the-art semi-supervised techniques.

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

# A    ADDITIONAL RESULTS

## A.1    RESULTS ON VTAB

We report the results of our VTAB experiment in Table 4. On VTAB, We find that both MULTI-SOURCE and SINGLE-SOURCE distillation outperforms IN+TRANSFER on each of the • *Natural* tasks. Particularly, MULTI-SOURCE outperform IN+TRANSFER with 13.9%-points on CIFAR-10 and 10.6%-points on Sun397 and averaged across • *Natural* MULTI-SOURCE outperforms IN+TRANSFER with 5.1%-points. Average over • *Specialized* both MULTI-SOURCE and SINGLE-SOURCE outperforms IN+TRANSFER, although with a small margin. Finally, averaged over • *Structured* IN+TRANSFER outperform our methods, but due to the nature of these tasks, we do not expect source models to transfer well to these tasks.[2] Yet, we still obtain the best accuracy on DMLab, dSpr-Loc, and sNORB-Azimuth.

Table 4: Top-1 accuracy by dataset in VTAB. The accuracy for each task is in grey, and the average accuracy for each category of tasks is in black. Note, the • *Mean* is the average across all tasks, not categories. The largest value in each column is marked in bold. Here MULTI-SOURCE is with $p = 9$.

| | Caltech101 | CIFAR-100 | DTD | Flowers102 | Pets | SVHN | Sun397 | Natural | Camelyon | EuroSAT | Resisc45 | Retinopathy | Specialized | Clevr-Count | Clevr-Dist | DMLab | KITTI-Dist | dSpr-Loc | dSpr-Ori | sNORB-Azim | sNORB-Elev | Structured | Mean |
|---|---|---|---|---|---|---|---|---|---|---|---|---|---|---|---|---|---|---|---|---|---|---|---|
| IN+Transfer | 88.1 | 47.0 | 57.4 | 85.8 | 82.8 | 75.3 | 27.8 | 66.3 | 81.0 | 95.0 | 80.0 | 72.7 | 82.2 | 73.1 | 55.9 | 43.6 | 75.7 | 18.7 | 58.6 | 21.2 | 46.0 | 49.1 | 62.4 |
| Multi-Source | 88.6 | 60.9 | 62.4 | 86.1 | 84.4 | 79.0 | 38.4 | 71.4 | 80.6 | 95.9 | 83.3 | 72.2 | 83.0 | 57.4 | 45.6 | 44.6 | 67.7 | 27.4 | 44.9 | 23.9 | 38.2 | 43.7 | 62.2 |
| Single-Source | 88.9 | 59.5 | 61.9 | 86.2 | 84.5 | 79.5 | 37.6 | 71.1 | 80.5 | 95.8 | 83.2 | 71.7 | 82.8 | 60.5 | 45.4 | 45.2 | 67.9 | 20.8 | 40.6 | 24.2 | 36.5 | 42.6 | 61.6 |

## A.2    RELATIVE ACCURACY OF SINGLE-SOURCE DISTILLATION

Similarly to Table 3, we extend on our evaluation of how well the task similarity selects the best source models for single-source distillation. We report the ratio between the average test accuracy of the top-$k$ target models ranked using the task similarity and the average test accuracy for the actual top-$k$ target models found after the fact in Table 5, Table 3, and Table 7 for $k = 1$, $k = 3$, and $k = 5$, respectively.

We find that generally, using task similarity on feature representations rather than the corresponding pseudo-labels yields better rankings, but also that PARC show very little difference between features and pseudo-labels for all considered $k \in \{1, 3, 5\}$.

**Relative accuracy over all $k$.** The relative accuracy measure reported above is sensitive to $k$ and the actual accuracy values of the models. I.e. if a metric flips the order of the best and second best model when there is a notable performance gap between the two models, the relative accuracy for $k = 1$ will be low, and we might be mistaken to believe the metric is not working well. However, the metric might rank every model for $k > 2$ perfectly correct, and since we typically utilize the full set of source models, the initial mistake should not be detrimental to the selection of task similarity metric. Thus, in Figure 8 we plot the relative accuracy for each task similarity metric and all $k \in \{1, \ldots, S\}$. We find that while PARC on feature representations is outperformed by both PARC and CKA on pseudo-labels for $k < 3$, PARC on feature representations outperforms all the other metrics for $k \geq 3$. In particular, from Table 8 we have that on average over all $k < S$, PARC, performs the best.

## A.3    ABLATION OF $p$ FOR TASK SIMILARITY-WEIGHTED MULTI-SOURCE DISTILLATION

We report the values associated with Figure 7 for each target task and all considered choices of $p$ in Table 9.

---

[2]The • *Structured* tasks are mainly (ordinal) regression tasks transformed into classification tasks, and thus it seems reasonable to expect very general features (such as those from an ImageNet pre-trained model) to generalize better to such constructed tasks than specialized source models.

Table 5: Relative accuracy of top-1 single-source distilled target model selected by task similarity over the best model found in hindsight. We compute the test accuracy of the highest ranked target model (ranked by some task similarity) and divide this with the test accuracy of the performing target model.

| | | CIFAR-10 | CUB200 | ChestX | EuroSAT | ISIC | NABird | Oxford Pets | Stanford Dogs | Mean |
|---|---|---|---|---|---|---|---|---|---|---|
| Pseudo | CKA | **99.6** | **100.0** | **96.1** | 99.5 | 98.1 | **100.0** | **100.0** | **100.0** | **99.2** |
| | PARC | 99.3 | **100.0** | 93.6 | 99.5 | **98.3** | **100.0** | 98.4 | **100.0** | 98.6 |
| | RSA | 99.3 | 74.8 | 94.8 | 99.5 | **98.3** | 86.6 | 97.8 | 95.6 | 93.4 |
| Feature | CKA | **99.6** | 81.0 | 92.6 | **99.8** | **98.3** | **100.0** | **100.0** | **100.0** | 96.4 |
| | PARC | **99.6** | **100.0** | 94.6 | 99.5 | 97.7 | **100.0** | 95.1 | **100.0** | 98.3 |
| | RSA | **99.6** | **100.0** | 92.6 | 99.5 | **98.3** | 80.6 | **100.0** | **100.0** | 96.3 |

Table 6: (Identical to Table 3) Relative accuracy of top-3 single-source distilled target models selected by task similarity over the average of the 3 best models found in hindsight. We compute the average test accuracy of the top-3 highest ranked target models and divide this average with the average test accuracy of the 3 best performing target models.

| | | CIFAR-10 | CUB200 | ChestX | EuroSAT | ISIC | NABird | Oxford Pets | Stanford Dogs | Mean |
|---|---|---|---|---|---|---|---|---|---|---|
| Pseudo | CKA | 99.1 | 95.6 | 97.4 | 99.6 | 98.8 | 89.4 | **100.0** | 97.6 | 97.2 |
| | PARC | 99.5 | **100.0** | 95.5 | 99.6 | 98.5 | 99.7 | 98.8 | **99.7** | 98.9 |
| | RSA | **100.0** | 77.7 | 96.5 | 99.7 | 98.5 | 87.2 | 98.6 | 97.6 | 94.5 |
| Feature | CKA | **100.0** | 95.6 | 97.0 | **99.8** | **99.0** | 93.3 | **100.0** | 96.4 | 97.6 |
| | PARC | **100.0** | **100.0** | **97.8** | 99.7 | 98.3 | **100.0** | 97.1 | 98.5 | **98.9** |
| | RSA | **100.0** | **100.0** | 96.7 | 99.8 | 98.9 | 94.9 | 98.9 | 98.8 | 98.5 |

Table 7: Relative accuracy of top-5 single-source distilled target models selected by task similarity over the average of the 5 best models found in hindsight. We compute the results analogously to Table 3 with $k = 5$.

| | | CIFAR-10 | CUB200 | ChestX | EuroSAT | ISIC | NABird | Oxford Pets | Stanford Dogs | Mean |
|---|---|---|---|---|---|---|---|---|---|---|
| Pseudo | CKA | 99.3 | 98.7 | **98.3** | 99.7 | 99.0 | 92.9 | 99.2 | 98.4 | 98.2 |
| | PARC | 99.7 | **100.0** | 96.7 | 99.7 | 98.9 | 94.5 | **99.4** | 98.4 | 98.4 |
| | RSA | 99.7 | 83.2 | 97.6 | 99.8 | 99.0 | 84.9 | 99.2 | 92.8 | 94.5 |
| Feature | CKA | **99.7** | 97.4 | 97.7 | 99.8 | 98.9 | 96.5 | 99.2 | 97.8 | 98.4 |
| | PARC | **99.7** | 100.0 | 97.9 | 99.8 | 99.1 | **99.7** | 97.5 | **99.7** | 99.2 |
| | RSA | **99.7** | 99.7 | 97.9 | **99.8** | **99.2** | 97.9 | 98.9 | **99.7** | 99.1 |

Table 8: The mean relative accuracy, across all $k$, for each metric in Figure 8. The average is bounded in $(0, 1]$, and 1 corresponds to perfect ordering by task similarity. We find that using feature representations consistently outperforms pseudo-labels and that for both feature representations and pseudo-labels PARC performs the best.

| | CKA | PARC | RSA |
|---|---|---|---|
| Pseudo | 0.985 | 0.990 | 0.974 |
| Feature | 0.986 | **0.993** | 0.991 |

## A.4 MULTI-SOURCE DISTILLATION WITH RESNET-50 AS TARGET ARCHITECTURE

In the main part of the article, we consider the computationally constrained setting, where some compute budget restricts the possible size of our target model. Thus, we use MobileNetV3 models as target models throughout the main paper. However, in Table 10 we remove the computational budget and allow the target model to be of any architecture, and particularly we use a ResNet-50 as the target model.

We compare multi-source distillation (with $p = 0$ and $p = 12$) initialized with either ImageNet pre-trained weights or the weights of the highest ranked ResNet-50 source model to TRANSFER and FINE-TUNE SELECTED SOURCE. We find that multi-source distillation initialized from ImageNet outperform TRANSFER on average for both equal weighting and $p = 12$, but underperform FINE-TUNE SELECTED SOURCE for both $p$. However, since FINE-TUNE SELECTED SOURCE is initialized from well-selected source model weights, the comparison is not entirely fair. Thus, we also consider the case where we initialize the target model for multi-source distillation with the weights of the highest ranked ResNet-50 source model, and find that for $p = 12$ multi-source distillation performs on par with FINE-TUNE SELECTED SOURCE.

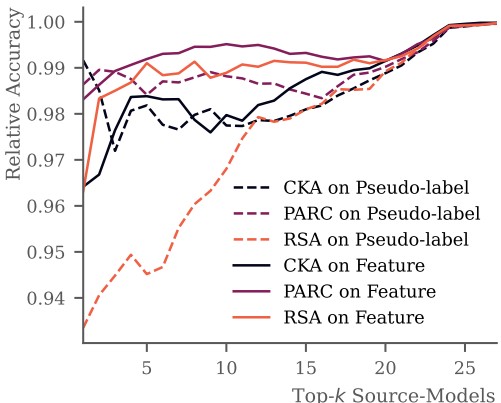

Figure 8: Relative accuracy of top-$k$ single-source distilled target models selected by task similarity over the average of the top-$k$ actual best target models found in hindsight. If the ordering by task similarity is perfectly correct, the relative accuracy would be 1 for all $k$. See Table 8 for the average of each metric across all $k$.

Table 9: Test accuracy of multi-source distillation with various choices of $p$, compared to the baseline methods of IN+TRANSFER and IN+FIXMATCH. We highlight the largest value for each target task in **bold**, and the results are also visualised in Figure 7.

| | CIFAR-10 | CUB200 | ChestX | EuroSAT | ISIC | NABird | Oxford Pets | Stanford Dogs | Mean |
|---|---|---|---|---|---|---|---|---|---|
| IN+FixMatch | **93.5** | 41.9 | 38.5 | **98.1** | **82.6** | 42.8 | 83.4 | 65.8 | 68.3 |
| IN+Transfer | 92.4 | 42.8 | 47.3 | 97.4 | 81.6 | 37.3 | 75.9 | 62.6 | 67.2 |
| Multi-Source (Equal) | 90.8 | 53.5 | 45.7 | 97.5 | 81.5 | 41.4 | 82.1 | 62.1 | 69.3 |
| Multi-Source ($p = 1$) | 91.1 | 55.6 | 46.5 | 97.9 | 81.5 | 42.5 | 83.3 | 64.4 | 70.3 |
| Multi-Source ($p = 3$) | 91.6 | 57.7 | 46.5 | 97.7 | 82.3 | 44.5 | 84.6 | 67.4 | 71.6 |
| Multi-Source ($p = 6$) | 91.8 | 59.0 | 46.7 | 97.5 | 82.5 | 46.7 | **84.7** | 69.1 | 72.3 |
| Multi-Source ($p = 9$) | 92.0 | 59.6 | 46.8 | 97.6 | 82.4 | 47.6 | 84.5 | 69.5 | 72.5 |
| Multi-Source ($p = 12$) | 92.0 | 60.0 | **47.7** | 97.6 | 82.2 | **48.3** | 84.4 | 69.9 | **72.8** |
| Multi-Source ($p = 15$) | 92.6 | **60.3** | 46.7 | 97.5 | 81.7 | 48.2 | 83.9 | 70.2 | 72.6 |
| Single-Source | 92.0 | 59.6 | 46.8 | 97.4 | 81.0 | 47.4 | 81.9 | **71.3** | 72.2 |

## A.5 NORMALIZATION OF TASK SIMILARITY FOR SOURCE MODEL WEIGHTING

We propose to choose the weights $\boldsymbol{\alpha} = (\alpha_1, \ldots, \alpha_S)$ as

$$\alpha_i = \frac{\underline{e}_i^p}{\sum_{s=1}^S \underline{e}_s^p}, \quad \text{where} \quad \underline{e}_j = \mathbb{1}_{(e_j > 0)} \, e_j \quad \text{for } j = 1, \ldots, S,$$

and $e_s$ is the task similarity for source model $\mathcal{M}_s$, evaluated on the target task, normalized to satisfy $e_s \in [0, 1]$ with min-max normalization over all $e_s$. Here, the hyperparameter, $p$ can be used to increase/decrease the relative weight on the highest ranked source models, with the extremes $p = 0$ and $p \to \infty$ corresponding to equal weight and single-source distillation, respectively. An alternative way to obtain our normalization is to use the softmax function on the task similarities,

$$\alpha_i = \frac{\exp\left(\frac{e_i}{T}\right)}{\sum_{s=1}^S \exp\left(\frac{e_s}{T}\right)}.$$

This does not require clipping the task similarity at $0$, and with the temperature, $T$, we can adjust the relative weight on particular source models. Here, large $T$ flattens the weights, and $T \to \infty$ corresponds to an equal weighting of all source models, while small $T$ increases the weight on the highest ranked source models. Quantitatively, the two normalization methods can yield similar transformations with appropriate choices of $p$ and $T$ - see Figure 9.

Table 10: Multi-Source Distillation with ResNet-50 as target model architecture. We compare fine-tuning of the highest ranked source model (Bolya et al., 2021) with multi-source distillation to both ImageNet-initialized target models and target models initialized from the highest ranked ResNet-50 source model. For $p = 12$, multi-source distillation performs on par with fine-tuning the selected source model. Largest value for each target tasks is in **bold**.

| | Target Model Initialization | CIFAR-10 | CUB200 | ChestX | EuroSAT | ISIC | NABird | Oxford Pets | Stanford Dogs | Mean |
|---|---|---|---|---|---|---|---|---|---|---|
| Transfer | ImageNet | 92.9 | 42.0 | 43.4 | 96.8 | 79.9 | 39.9 | 83.3 | 65.9 | 68.0 |
| Fine-tune Selected Source | Source Model | **93.0** | 70.8 | 43.9 | 97.2 | **81.3** | 47.4 | 84.8 | **79.3** | **74.7** |
| Multi-Source (Equal) | ImageNet | 87.8 | 57.3 | 46.1 | 97.0 | 78.9 | 42.4 | 84.1 | 64.5 | 69.8 |
| Multi-Source ($p = 12$) | ImageNet | 91.5 | 64.5 | 45.4 | 97.0 | 78.9 | 49.8 | **87.1** | 74.2 | 73.6 |
| Multi-Source (Equal) | Source Model | 87.5 | 68.8 | 45.5 | **97.4** | 81.2 | 43.2 | 81.9 | 65.1 | 71.3 |
| Multi-Source ($p = 12$) | Source Model | 91.6 | 70.0 | **47.6** | 97.0 | 80.8 | **50.0** | 85.7 | 73.8 | 74.6 |

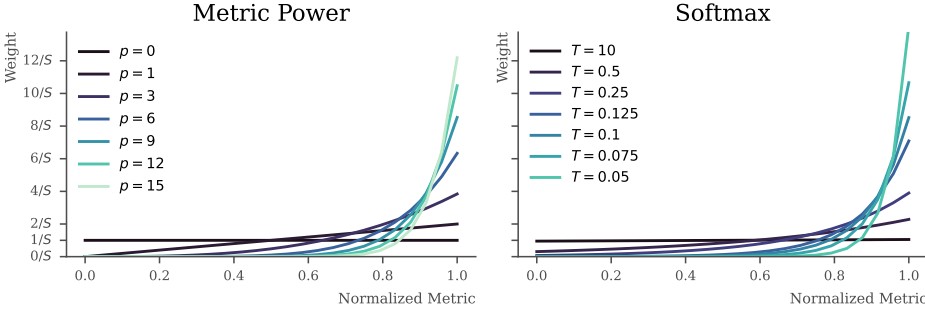

Figure 9: Transformation of weights for various choices of power (left) or softmax temperature (right). Here $S$ is the number of source models, and we consider equidistantly distributed normalized metrics.

Table 11: Multi-Source Distillation on a variation of SDMS with only 5% labeled samples per task. Again, we compare to the baseline of IN+TRANSFER. Largest value for each target tasks is in **bold**.

| | CIFAR-10 | CUB200 | ChestX | EuroSAT | ISIC | NABird | Oxford Pets | Stanford Dogs | Mean |
|---|---|---|---|---|---|---|---|---|---|
| IN+Transfer | 88.0 | 16.8 | **43.5** | 94.8 | 73.9 | 14.4 | 55.0 | 38.9 | 53.2 |
| Multi-Source ($p = 1$) | 88.1 | 29.2 | 42.3 | **95.9** | 76.3 | 20.5 | 66.6 | 42.1 | 57.6 |
| Multi-Source ($p = 9$) | **90.2** | **32.3** | 42.6 | 95.9 | **76.7** | **24.8** | **68.2** | 49.0 | **60.0** |
| Single-Source | 87.2 | 31.4 | 39.7 | 95.1 | 75.4 | 24.0 | 58.9 | **49.7** | 57.7 |

## A.6 SMALLER AMOUNT OF LABELED DATA

We now repeat the experiment of the main paper across the 8 target datasets with a reduced amount of labeled samples. Here, we reduce the number of labeled samples to $5\%$ (rather than $20\%$) of the training set and report the accuracy in Table 11. We find a similar pattern as observed in the main experiment, where MULTI-SOURCE distillation on average outperforms TRANSFER irrespective of the choice of $p$. For $p = 9$ MULTI-SOURCE outperforms TRANSFER by $6.8\%$-point on average and in particular $15.5\%$-points on CUB200, whereas the only loss in performance is on ChestX with a drop of $0.9\%$-point.

Table 12: Pearson correlation between test accuracy after all possible single-source distillations and task similarity associated with the source models. Similar to Table 2.

| | | CIFAR-10 | CUB200 | ChestX | EuroSAT | ISIC | NABird | Oxford Pets | Stanford Dogs | Mean |
|---|---|---|---|---|---|---|---|---|---|---|
| Pseudo | CKA | 0.62 | **0.85** | 0.07 | 0.30 | -0.06 | 0.33 | 0.67 | 0.21 | 0.37 |
| | PARC | 0.75 | 0.74 | -0.03 | 0.27 | -0.00 | 0.36 | 0.63 | 0.51 | 0.40 |
| | RSA | 0.75 | 0.13 | -0.07 | 0.38 | 0.04 | -0.09 | 0.66 | 0.40 | 0.27 |
| Feature | CKA | 0.84 | 0.60 | **0.39** | 0.29 | 0.00 | 0.30 | 0.71 | 0.54 | 0.46 |
| | PARC | 0.86 | 0.73 | 0.17 | **0.46** | -0.06 | **0.58** | 0.77 | 0.78 | **0.54** |
| | RSA | **0.90** | 0.85 | 0.07 | 0.45 | **0.04** | 0.27 | **0.87** | **0.83** | 0.54 |

Table 13: Kendall Tau correlation between test accuracy after all possible single-source distillations and task similarity associated with the source models. Similar to Table 2.

| | | CIFAR-10 | CUB200 | ChestX | EuroSAT | ISIC | NABird | Oxford Pets | Stanford Dogs | Mean |
|---|---|---|---|---|---|---|---|---|---|---|
| Pseudo | CKA | 0.51 | 0.46 | 0.16 | 0.28 | -0.05 | 0.24 | 0.49 | 0.07 | 0.27 |
| | PARC | 0.61 | 0.64 | 0.01 | 0.12 | 0.02 | 0.36 | 0.54 | 0.39 | 0.34 |
| | RSA | 0.62 | 0.17 | -0.07 | 0.22 | **0.08** | -0.01 | 0.48 | 0.29 | 0.22 |
| Feature | CKA | 0.67 | 0.34 | **0.25** | 0.14 | -0.05 | 0.40 | 0.50 | 0.38 | 0.33 |
| | PARC | 0.69 | **0.67** | 0.14 | **0.31** | -0.10 | **0.65** | 0.62 | 0.67 | **0.46** |
| | RSA | **0.72** | 0.65 | 0.02 | 0.28 | 0.02 | 0.19 | **0.72** | **0.67** | 0.41 |

## A.7 DIFFERENT MEASURES OF CORRELATION

In order to evaluate the quality of a task similarity metric to estimate the performance of a target model after distillation, we consider the correlation between the computed metric and the actual observed performance after distillation. However, since we have no reason to believe that the relationship is linear, we consider the Spearman correlation in the main paper. However, for completeness of exposition we report Pearson correlation and Kendall's Tau in Table 12 and Table 13, respectively. For both these correlation measures, the overall conclusions are the same: Using feature representations is preferable to pseudo-labels, and PARC generally outperforms both CKA and RSA, albeit not by much over CKA.

## A.8 CHOICE OF TASK SIMILARITY METRICS

Recently, multiple measures intended to estimate the transferability of a source model have been proposed. However, despite the very recently published Multi-Source Leep (MS-LEEP) and Ensemble Leep (E-Leep) no task similarity metric consider the estimation over multiple models at once (Agostinelli et al., 2022). Thus, we consider each source model separately and compute the metrics independent of other source models. This has the added benefit of reducing the number of metric computations required as we do not need to compute the task similarity for all possible combinations of $n$ models from $S$ possible (i.e. $\binom{n}{S}$), which grows fast with $S$.

Assume $\mathbf{X} \in \mathbb{R}^{N \times d_X}$ and $\mathbf{Y} \in \mathbb{R}^{N \times d_Y}$, and that $\mathbf{K}_{ij} = k(\mathbf{x}_i, \mathbf{x}_j)$ for and $\mathbf{L}_{ij} = l(\mathbf{y}_i, \mathbf{y}_j)$ where $k$, and $l$ are two (similarity) kernels as well as $\mathbf{x}_i, \mathbf{x}_j$ and $\mathbf{y}_i, \mathbf{y}_j$ are rows of $\mathbf{X}$ and $\mathbf{Y}$, respectively. Then we have that CKA is defined as

$$\rho_{\text{CKA}}(\mathbf{X}, \mathbf{Y}) \stackrel{\text{def}}{=} \frac{\text{HSIC}(\mathbf{K}, \mathbf{L})}{\sqrt{\text{HSIC}(\mathbf{K}, \mathbf{K})\text{HSIC}(\mathbf{L}, \mathbf{L})}},$$

where $\mathbf{K}, \mathbf{L} \in \mathbb{R}^{N \times N}$ and HSIC is the Hilbert-Schmidt Independence Criterion,

$$\text{HSIC}(\mathbf{K}, \mathbf{L}) \stackrel{\text{def}}{=} \frac{1}{(N-1)^2} \text{tr}\left(\mathbf{K}\mathbf{H}_N\mathbf{L}\mathbf{H}_N\right), \quad \text{with } \mathbf{H}_N \stackrel{\text{def}}{=} \mathbf{I}_N - \frac{1}{N}\mathbf{1}\mathbf{1}^\top.$$

In particular, if both $k$ and $l$ are linear kernels, then we have that

$$\rho_{\text{CKA}}(\mathbf{X}, \mathbf{Y}) = \frac{\|\mathbf{Y}^\top\mathbf{X}\|_F^2}{\|\mathbf{X}^\top\mathbf{X}\|_F\|\mathbf{Y}^\top\mathbf{Y}\|_F},$$

where $\|\cdot\|_F$ is the Frobenius norm. We use the linear kernel throughout this paper and refer to Cortes et al. (2012) for additional details on CKA.

For RSA, we consider the dissimilarity matrices given by

$$\mathbf{K}_{ij} \stackrel{\text{def}}{=} 1 - \text{corrcoef}(\mathbf{x}_i, \mathbf{x}_j) \quad \text{and} \quad \mathbf{L}_{ij} \stackrel{\text{def}}{=} 1 - \text{corrcoef}(\mathbf{y}_i, \mathbf{y}_j),$$

where $\mathbf{X}$ and $\mathbf{Y}$ are assumed normalized to have mean $0$ and variance $1$. We then compute RSA as the Spearman correlation between the lower triangles of $\mathbf{K}$ and $\mathbf{L}$,

$$\rho_{\text{RSA}}(\mathbf{X}, \mathbf{Y}) \stackrel{\text{def}}{=} \text{spearman}\left(\{\mathbf{K}_{ij} \mid i < j\}, \{\mathbf{L}_{ij} \mid i < j\}\right).$$

For additional details on RSA, we refer the reader to Dwivedi & Roig (2019). While Bolya et al. (2021) introduces PARC alongside a heuristic and feature reduction, the PARC metric is almost identical to RSA. However, RSA was introduced to compute similarities between two sets of representations, and PARC was aimed at computing similarities between a set of representations and a set of labels associated with the dataset. Thus, in our use of PARC, it merely differs from RSA in the lack of normalization of $\mathbf{Y}$, which is assumed to be one-hot encoded vectors of class labels from the probe dataset.

## B  EXPERIMENTAL DETAILS

In the following, we provide some experimental details.

### B.1  MAIN EXPERIMENTS

Unless otherwise mentioned, we use SGD with a learning rate of $0.01$, weight decay of $0.0001$, batch size of $128$, and loss weighting of $\lambda = 0.8$. We initialize our target models with the ImageNet pre-trained weights available in torchvision (`https://pytorch.org/vision/stable/models`), and consider the 31 fine-tuned models from Bolya et al. (2021) publicly available at `github.com/dbolya/parc` as our set of source models. The source models consist of each of the architectures (AlexNet, GoogLeNet, ResNet-18, and ResNet-50) trained on CIFAR-10, Caltech101, CUB200, NABird, Oxford Pets, Stanford Dogs, VOC2007, and ImageNet.[3] Note, we always exclude any source model trained on the particular target task, thus effectively reducing the number of source models for some target tasks. For FixMatch we use a batch size of $128$ (with a 1:1 ratio of labeled to unlabeled samples for each batch) and fix the confidence threshold at $0.95$ and the temperature at $1$. We keep the loss weighting between the supervised loss and the unlabeled FixMatch loss at $\lambda = 0.8$.

### B.2  VTAB EXPERIMENTS

For each VTAB experiment we consider the full training set (as introduced in Zhai et al. (2019)) as the unlabeled set, $\mathcal{D}_\tau^u$, and the VTAB-1K subset as the labeled set, $\mathcal{D}_\tau^l$. We use the Pytorch implementation from Jia et al. (2022) available at `github.com/KMnP/vpt`.

We use SGD with a learning rate of $0.005$, weight decay of $0.0001$, batch size of $128$ equally split in $64$ labeled and unlabeled samples, and loss weighting of $\lambda = 0.9$. We train our models for $100$ epochs, where we define one epoch as the number of steps required to traverse the set of unlabeled target data, $\mathcal{D}_\tau^u$ when using semi-supervised methods, or merely as the number of steps to traverse the labeled set, $\mathcal{D}_\tau^l$, for supervised transfer methods. We initialize our target models with the BiT-M ResNet-50x1 model fine-tuned on ILSVRC-2012 from BiT (Kolesnikov et al., 2020) publicly available at `github.com/google-research/big_transfer`.

We consider the 19 BiT-M ResNet-50x1 models fine-tuned on the VTAB-1K target tasks from Kolesnikov et al. (2020) as the set of source models. We always exclude the source model associated with the target task from the set of source models, and thus effectively have 18 source models available for each target task in VTAB. We use the PARC metric on the source model features to compute the source weighting, but also only use the top-5 highest ranked source models to reduce the computational costs of training. Furthermore, we use $p = 9$ for multi-source distillation.

---

[3]Note, there is no ResNet-50 trained on ImageNet in the set of source models, thus the total number of models is 31 and not 32 as otherwise expected.

