# OpenReview forum: "Learning Efficient Models From Few Labels By Distillation From Multiple Tasks"
_ICLR.cc/2023/Conference — Submitted to ICLR 2023_

### Official Review · Reviewer_fFzJ · 2022-10-24

**Confidence:** 4
**Correctness:** 2
**Technical Novelty And Significance:** 2
**Empirical Novelty And Significance:** 2
**Recommendation:** 5

**Clarity, Quality, Novelty And Reproducibility:**

The paper is well written and complete in terms of reproducibility. Although some findings are interesting, they are not completely surprising. In particular Single Source is quite similar to the proposed one. Also, some backbones are weak: InceptionV3, InceptionResNet and others could be used as usually those transfer better (Kornblith 2019), which puts into question the claims of "distillation is better than fine-tuning from ImageNet" and "distilling to efficient architecture could be better than fine-tuning large models".

**Strength And Weaknesses:**

Strengths:
- The paper shows there are sources that may represent better some given target tasks than others, including the baseline of ImageNet pretrained networks
- The methods show superior results in particular for fine-grained data such as CUB200

Weaknesses:
- FixMatch is a semi-supervised technique where the unlabeled data is assumed to be from the same domain
- The methods assumes the target task is available during distillation time
- A stronger baseline would be ViT/DINO
- It seems that, overall, it would be better to select a few tasks (more relevant) than to weigh them all. Maybe a preselection would reach similar results without computational burden (single is almost as good as task similarity)
- It is interesting that MobileNet performs better than ResNet (among others), which contradicts previous work, and no justification is given.

**Summary Of The Paper:**

The paper assumes limited label availability and propose a weighted multi-source distillation method, in practice that means distill multiple (diverse) source models trained on different domains, weighing them by their relevance for the target task, assuming such target task is available

**Summary Of The Review:**

The paper is interesting overall but there are better backbones available, including transformed-based ones. Also, single source is quite similar to the proposed method.

---

### Official Review · Reviewer_Tp8o · 2022-10-25

**Confidence:** 4
**Correctness:** 3
**Technical Novelty And Significance:** 3
**Empirical Novelty And Significance:** 3
**Recommendation:** 5

**Clarity, Quality, Novelty And Reproducibility:**

The idea of this paper is interesting and the paper is well-written. But there is not too much technical contribution presented in this paper. The task relevance metric is important for the proposed method, but the authors didn't discuss this issue.


**Strength And Weaknesses:**

*Strength
1. The idea of distiallting knowledge from multiple datasets with weights is interesting and reasonable. The authors conduct a lot of experiments to validate the corresponding claims.
2. The paper is overall clear and well-organized in illustrating the methods, motivation, and experimental results.

*Weakness
1. The task relevance metric is very important for knowledge distillation, but the authors don't discuss the influence of the different choices of the task relevance metric.

**Summary Of The Paper:**

This paper address the challenge of knowledge distillation for image recognition tasks with few training samples. This paper proved that simply distilling knowledge from a single task will not get good performance. The authors present a weighted multi-source distillation method to distill multiple source models trained on different domains by weighting these models with a task relevance score. Experiments demonstrate that more datasets will bring performance improvement.

**Summary Of The Review:**

I think the method and experiments are not strong enough to support the motivation of this paper. Importance issues are forgotten to be discussed.

---

### Official Review · Reviewer_QfTL · 2022-10-26

**Confidence:** 3
**Correctness:** 3
**Technical Novelty And Significance:** 3
**Empirical Novelty And Significance:** 2
**Recommendation:** 5

**Clarity, Quality, Novelty And Reproducibility:**

The method overall is novel but the technical contribution and empirical results are not significant enough.

**Strength And Weaknesses:**

Strength:
- Learning with limited labeled data is meaningful to be solved.
- The experiments and ablation studies are well-designed.

Weakness:
- In the main table it's good to show how multi-source distillation works for all architectures.
- The empirical results are not convincing enough comparing to the single-source distillation with task similarity. Seems it only provides marginal improvement with multi-source but adds way more computation overhead.
- Extra hyperparameters are required to be tuned (especially p).
- I don't quite get why this method especially targets efficient models and doesn't work well for large models like resnet50. More analyzes will be very helpful.

**Summary Of The Paper:**

In this paper, the authors propose a method to train efficient models with limited labeled data. By using task similarity metrics, they conduct weighted knowledge distillation with pretrained models from different sources. It achieves better results than transferring from ImageNet pretrained model or FixMatch.

**Summary Of The Review:**

In conclusion, this work shows task similarity is useful metrics when it comes to distillation with teachers from multiple sources. I think more evidences are needed to prove multi-source distillation is indeed better than single-source distillation.

---

### Decision · Program_Chairs · 2023-01-20

**Decision:**

Reject

**Justification For Why Not Higher Score:**

While the method is simple, and the paper is clearly written and well motivated, all reviewers and the AC agree that the paper is marginally below of the ICLR acceptance threshold, due to incremental novelty, minor gains of multi-distillation, issues with generalization, clarity about source datasets, and other concerns related to backbones and task similarity metrics.

**Justification For Why Not Lower Score:**

N/A

**Metareview: Summary, Strengths And Weaknesses:**

This paper presents a method based on distillation from multiple pretrained models for efficient visual learning with limited labels in a semi-supervised setting. All three reviewers consider the paper is marginally below the ICLR acceptance threshold. Overall, the proposed approach is simple, which is a nice property of this work. The paper is also very well written and clearly motivated. In the AC-reviewers meeting, we felt the novelty of the method is not high, as other previous methods, such as "Knowledge fusion" (ICLR 2019), share similar ideas of multi-source distillation with disjoint tasks, although not done in a semi-supervised setting. The reviewers also pointed out that the gains of multi- versus single-distillation are very minor. In the rebuttal, the authors claim that single-distillation is a contribution of their work, but the paper story/motivation is centered around multi-source (instead of single-source) distillation. It could be better to use more complementary source models, like attributes, which could be combined to better showcase the distillation from multiple models. Related to that, we felt the paper does not clearly convey the list of source datasets, as described in previous work. This would facilitate the understanding of the obtained gains. The proposed method does not seem to generalize well to different domains, as shown in the VTAB results. Finally, the reviewers also raised other concerns related to backbones and task similarity, which were not eliminated after the author response. The AC agrees with the reviewers and encourages the authors to improve the paper for another top conference submission.

**Summary Of Ac-Reviewer Meeting:**

We have discussed the author response, which was well received, but unfortunately the consensus was that the paper remains below the acceptance threshold of ICLR.

The discussion included the novelty of the work, the single vs multi-source distillation results, the list of source datasets, generalization to other domains, different backbones, and task similarity metrics. See the meta-review above for the comments about these points.